# Comparative Study on UV Degradation of Black Chinese Lacquers with Different Additives

**DOI:** 10.3390/ma16165607

**Published:** 2023-08-13

**Authors:** Wenjia Liu, Xinyou Liu, Jiufang Lv

**Affiliations:** 1Co-Innovation Center of Efficient Processing and Utilization of Forest Resources, Nanjing Forestry University, Nanjing 210037, China; liuwenjiajia@hotmail.com; 2College of Furnishing and Industrial Design, Nanjing Forestry University, Str. Longpan No. 159, Nanjing 210037, China; 3Faculty of Furniture Design and Wood Engineering, Transilvania University of Brasov, 500036 Brasov, Romania

**Keywords:** UV degradation, Chinese lacquer, black lacquer, FTIR, XPS

## Abstract

This study investigates the UV degradation of black Chinese lacquer by incorporating carbon black and ferrous hydroxide as additives. The purpose of this research is to understand the effects of these additives on the degradation behavior of the lacquer film. Different concentrations of carbon black powder (1%, 3%, and 5%) and Fe(OH)_2_ (10%, 20%, and 30%) were added to the lacquer following traditional techniques. The main methods employed for analysis were gloss loss measurement, color change assessment, SEM imaging, FTIR spectroscopy, and XPS analysis. The results demonstrate a significant decrease in gloss levels and an increase in lightness values with increasing ultraviolet exposure time. SEM images reveal the formation of cracks in the lacquer film. FTIR analysis indicates oxidation of the urushiol side chain and an increase in oxidation products. The infrared difference spectrum highlights the differences between the additives, with Fe(OH)_2_ showing a lower impact on the spectra compared to carbon black. XPS analysis confirms the oxidation of the C-H functional group and the presence of C-O-C and C-OH groups. In conclusion, this study sheds light on the influence of carbon black and ferrous hydroxide additives on the UV degradation of black Chinese lacquer and suggests the protective effect of Fe(OH)_2_ against UV aging. These findings contribute to a better understanding of the degradation mechanisms and provide insights for improving the UV resistance of Chinese lacquer coatings. Further research can explore alternative additives and optimization strategies to mitigate UV-induced degradation.

## 1. Introduction

Chinese lacquer is a natural, biodegradable, renewable, and environmentally friendly coating [1,2] comprising main components such as urushiol, laccase, and polysaccharides. Among these, urushiol constitutes 40–80% of the total raw lacquer content, varying depending on its origin. The history of lacquer usage dates back more than 8000 years [3,4]. Known for its high hardness, glossiness, corrosion resistance, high temperature resistance, waterproof properties, and durability, traditional lacquerware has withstood the test of time and is preserved to this day. To enhance the aesthetic appeal of raw lacquer, ancient craftsmen added natural pigments to create various colors. 

Among the excavated artifacts, black lacquerware emerges as the most prevalent among the early finds. Notably, the earliest dated Jingtoushan lacquerware, dating back over 8000 years, was also black [4]. The black lacquer used in the screen discovered in the tomb of Sima Jinlong during the Northern Wei Dynasty (A.D. 386–A.D. 534) was identified as charcoal black [5]. Similarly, lacquerware from the Song Dynasty (A.D. 960–A.D. 1279), such as lacquer cups and lacquer plates, revealed traces of charcoal and iron, leading to the inference that the black lacquer’s composition derived from a combination of charcoal black and iron oxide [6]. The same charcoal black composition was confirmed in the surface lacquer layer of Emperor Qianlong’s coffin during the Qing Dynasty [7]. Additionally, the black hue in traditional Korean lacquerware was also attributed to its iron content [8].

A historical account of black lacquer production is found in the *Nan Cun Chuo Geng Lu南村辍耕录* by Tao Zongyi during the Yuan Dynasty (A.D. 1271–A.D. 1368). According to these records, Chinese lacquer was boiled, and black lacquer was obtained by soaking scrap iron filings in rice vinegar [9]. Modern scholars have come to believe that this process involves the following reaction [10]:CH_3_COOH + Fe_2_O_3_ → Fe(OH) (CH_3_COO)_2_ + H_2_OFe(OH) (CH_3_COO)_2_ + H_2_O → Fe (OH)_2_↓ + CH_3_COOHFe (OH)_2_ + O_2_ + H_2_O → Fe(OH)_3_↓

*Xiu shi lu髹饰录*, along with related studies, suggest that additives such as carbon black or ferrous hydroxide were commonly used in the preparation of black lacquer [11,12]. These historical records align well with the identification results of the actual excavated artifacts.

Over time, lacquerware may experience issues such as discoloration, fading, cracking, and deterioration due to environmental factors or inadequate protection measures. Previous studies have attributed these problems to the poor UV stability and the brittleness of raw lacquer, particularly under dry conditions [13,14]. The degradation of lacquer films is primarily caused by prolonged exposure to natural light [15,16]. Even visible light within the range of 510–650 nm wavelengths has a significant impact on lacquer film degradation [17]. UV radiation is especially damaging to lacquer films [18], although most of it is absorbed by the thin surface layer of the film [19]. With increased exposure to UV light, the hydroxyl group content in the lacquer film increases, the carbon peak decreases, and the oxygen content rises, indicating an accelerated chemical reaction due to UV exposure [20,21]. Both sunlight and visible light sources contribute to lacquer film damage, with the extent of damage directly proportional to light intensity [14], and cumulative damage escalating with time [22].

Existing research on black lacquer predominantly employs UV light as the primary light source for studying its degradation characteristics over a duration of 16–28 days [23]. During this period, black lacquer exhibits noticeable signs of aging. Common physical property characterization methods for black lacquer coatings include gloss, color, roughness, and contact angle measurements. Japanese scholars often employ gloss residual rate [20] to assess the UV degradation properties of black lacquer. Color difference tests utilize ΔL to indicate brightness. Mechanical performance tests are also conducted to study the viscoelastic response of lacquer coatings. Chemical property characterization primarily involves Fourier transform infrared spectroscopy (FTIR), X-ray photoelectron spectroscopy (XPS), thermogravimetric analysis coupled with mass spectrometry (TG-MS), and other techniques.

However, studies on the UV degradation of raw lacquer have primarily focused on the deterioration of its physical properties and chemical components under light conditions [24,25]. There has been limited in-depth research regarding the influence of black additives in black lacquer and the proportionate relationship of different components on the UV degradation performance of raw lacquer [26]. Hence, this study aims to conduct a comparative analysis of black lacquer formulations with varying iron and carbon black contents before and after UV degradation. The objective is to identify the optimal ratio of black lacquer additives to combat light-induced aging and gain insights into the effects of different black lacquer addition ratios on UV degradation performance. Furthermore, this research comparatively investigates the influence of additives on the UV aging mechanism of black lacquer.

## 2. Materials and Methods

### 2.1. Materials

The raw lacquer used in this study was sourced from a Chinese lacquer tree in Maoba, located in the Hubei province of China. It was purchased from a paint store in Nanjing. According to the test report provided by the merchant, the main components of the raw lacquer are urushiol, water, polysaccharides, glycoproteins, and laccase. Among these components, urushiol is a general term for polyphenolic compounds consisting of a mixture of catechol derivatives containing C_15_ and C_17_ sidechain aliphatic hydrocarbon substituents with 0 to 3 double bonds [27]. To enhance the curing speed and film quality of the raw lacquer, a refining process was conducted based on the literature [19,28], reducing the raw lacquer’s water content to 5%. Nanoscale carbon black powder was procured from a store in Nanjing, while ferrous hydroxide was obtained from the Coating Laboratory of Nanjing Forestry University. Additionally, pig blood material was purchased from Yangzhou Lacquer Factory, which was made from fresh pig blood with a small amount of lime and water, and stirred to produce gelatinous pig blood, used as a binder for the ground layer. 

Chinese fir wood (*Cunninghamia lanceolata (Lamb.) Hook.*) with a water content of 10 ± 2% was obtained from Zhejiang Jeson Wood Ltd. (Huzhou, China) The wood was cut into radial plates measuring 100 × 100 × 10 mm, resulting in a total of 48 pieces. These plates were sanded using 400-grit sandpaper and conditioned in a climatic chamber at (20 ± 2) °C and (55 ± 5)% relative humidity until a constant weight was reached (14 days), prior to sample preparation [29].

### 2.2. Preparation of Black Chinese Lacquer

The preparation of the black material in black lacquer is based on traditional documentation and archaeological findings. Traditional carbon black was directly purchased, while ferrous hydroxide in traditional iron black lacquer was prepared using modern methods commonly employed in the lacquer industry [30]. Nanoscale carbon black powder was added gradually to the Kurome lacquer at weight ratios of 1%, 3%, and 5% (0.83 mol/Kg, 2.50 mol/Kg, and 4.17 mol/Kg) and stirred thoroughly to obtain black Chinese lacquer. Ferrous hydroxide, prone to oxidation in the air and challenging to store, was prepared using ferrous sulfate and sodium bicarbonate through the following chemical reaction:FeSO_4_ + 2NaHCO_3_ = Fe(OH)_2_↓ + Na_2_SO_4_ + 2CO_2_↑

A uniform mixture of 1 mol FeSO_4_ solution and 2 mol 2NaHCO_3_ solution resulted in the formation of a precipitate. Filtration treatment was carried out to obtain finely pasted Fe(OH)_2_. Gradually, the Fe(OH)_2_ paste was added to the Kurome lacquer at weight ratios of 10%, 20%, and 30% (1.11 mol/Kg, 2.22 mol/Kg, and 3.33 mol/Kg) and stirred thoroughly to obtain black Chinese lacquer. In this process, the raw lacquer undergoes a chemical reaction with ferrous sulfate. The phenolic hydroxyl groups in the raw lacquer coordinate with the ferrous ions in ferrous sulfate, forming stable chemical bonds and resulting in the formation of the lacquer phenol–iron complex [31].

### 2.3. Sample Preparation

In the Chinese traditional finishing process, a ground layer is often applied between the Chinese lacquer and the wood to enhance the stability of the lacquer in the environment [32,33]. For this purpose, fresh pig blood was mixed with a small amount of lime and water to create colloidal pig blood, which served as an adhesive for the ground layer [34]. The jelly-like pig blood was combined with three different types of brick powder (coarse, medium, and fine) at a 1:1 ratio, resulting in three different types of putty. The prepared wood specimens were coated successively with coarse, medium, and fine putty layers. Each layer of putty was allowed to dry for 7 days, and the combined thickness of the three layers reached approximately 1 mm. Finally, the wood specimens were sanded with 400-grit sandpaper and finished with different black Chinese lacquer formulations. The specimens were then placed in a climatic chamber at 15–20 °C and 70–85% relative humidity for 21 days [35,36]. The black lacquer with different additives was uniformly applied to the surface of the ground layer coated with pig blood putty, resulting in 8 specimens per group, as shown in Table 1.

### 2.4. Accelerated Aging through Ultraviolet Irradiation

The processed specimens were placed in a UV aging test chamber (Hefei Anke Environmental Test Equipment Limited Company, Hefei, China, AKZN-P) for the UV aging test. The test followed the standard ASTM G154-16 [37] and utilized UVA340 nm lamp tubes with a temperature set at 60 °C. The specimens were exposed to UV irradiation at a rate of 0.89 W/(m^2^·nm). Samples were collected every 7 days, and data measurements were taken. The aging process consisted of 5 cycles, resulting in a total duration of 35 days.

### 2.5. Physicochemical Measurements

The glossiness of the black lacquer specimens during the UV aging process was measured using a Tri-angle Gloss Meter (Guangdong Sanenshi Intelligent Technology Co., Ltd., Shenzhen, China, HG268) in accordance with the ASTM D523-2014 Standard Test Method for Specular Gloss [38] and GB/T4893.6-2013 Physical and Chemical Properties Test of Furniture Surface Paint Film, Part 6 Gloss Measurement Method [39]. The gloss measurements were taken at an angle of 60°. Prior to starting the aging experiment and every 7 days during the aging process, glossiness data were measured at the same point on the same specimen. The gloss loss rate at the fixed point of each black lacquer film was calculated, and the average value was determined to obtain the gloss loss rate of different formulations of black lacquer.

Gloss loss formula:$$H=\frac{H_{0}-H_{i}}{H_{0}}\times 100\%$$

In the formula: *H*_0_: initial gloss, %; 

*H_i_*: gloss after 35 days of UV aging, %.

The color change in the samples was assessed using the ASTMD2244-2009b standard [40]. The CIEL*a*b* color space was employed, along with a D65 light source, 10° observation angle, and specular reflection inclusion conditions. The Pantone color detector color reader (Pantone, X-RITE, RM200-PT01, Bruker, Germany) was used for the color measurements. Color measurements were taken before the start of the aging experiment and every 7 days thereafter, at the same point on the same specimen each time. Since natural raw lacquer appears as dark brown after curing and the addition of ferrous hydroxide or carbon black makes the lacquer black, the values of Δ*a* and Δ*b* obtained from the test are very low. Therefore, in data analysis, Δ*L* is used to represent color lightness, and the changes in Δ*a* and Δ*b* are not further discussed. The lightness difference at the fixed point of each black lacquer film was calculated, and the average value was determined to obtain the lightness difference of different black lacquer films.

Lightness difference formula: Δ*L*= *L_i_* − *L*_0_.

In the formula: *L*_0_: initial color lightness; *L_i_*: color brightness after 35 days of UV aging.

The control and aged samples were coated and fixed on a metal support with conductive adhesive in a general room temperature environment (20 °C). After drying, platinum and gold plating were sprayed, and the morphological changes of the specimens during photoaging were observed using environmental scanning microscopy (FEI, Quanta 250, Hillsboro, OR, USA) under an accelerating voltage of 20 kV.

The surface functional groups of the control samples and the aged coatings were measured using an attenuated total reflectance FTIR spectrometer (Bruker, Mannheim, Germany, ALPHA-II) in the scan range of 4000 cm^−1^ to 500 cm^−1^ with an average of 32 scans.

The qualitative and quantitative analysis of surface elements of the control and aged coatings was completed using XPS (Shimadzu, AXIS UltraDLD, Kyoto, Japan). The cured coating was fixed on a double-sided adhesive tape. XPS spectroscopy was performed in the constant energy analyzer mode. The control and aged samples were scanned in full spectrum, analyzing individual elements in the 0–1200 eV range with a focus on the carbon spectrum.

### 2.6. Statistical Analysis

Statistical analysis was conducted using SPSS (IBM Corp., IBM SPSS Statistics for Windows, v. 25, Armonk, NY, USA). An ANOVA analysis was performed on the measured glossiness and lightness in the color measurement at a significance level of 0.05 to evaluate the significant effect of the amount of added dose on gloss and color during UV degradation. The Levene test was used to analyze the homogeneity of variance, and the Shapiro–Wilk test was employed to check the normal distribution of data, as both tests are prerequisites for one-way ANOVA tests. The mean values were separated using the Fisher protected minimum significant difference (LSD) test at α = 0.05.

## 3. Results and Discussion

### 3.1. Gloss Loss

Gloss measurement is a fundamental method for evaluating the surface of lacquer films [41]. Table 2 presents the gloss values of Chinese black lacquer with added Fe(OH)_2_ and carbon before and during UV aging. The addition of Fe(OH)_2_ has a noticeable effect on the gloss of the lacquer film, with the gloss increasing significantly as the amount of Fe(OH)_2_ added increases. In contrast, the glossiness of Chinese lacquer decreases significantly when the amount of carbon black added is 3–5% compared to 1%. However, there is no significant difference in glossiness between 3% and 5% carbon black additions. As the UV aging time increases, the glossiness of both types of black Chinese lacquer decreases significantly. After 35 days of UV aging, the black Chinese lacquer with 10% Fe(OH)_2_ added exhibits the lowest glossiness, reaching 1.81%. 

Figure 1 illustrates the trend of glossiness loss in black lacquers with different additives. The maximum loss in glossiness occurs during the first UV aging cycle, with a gradual decrease observed in subsequent cycles. During UV aging, the black Chinese lacquer with 10% Fe(OH)_2_ additive experiences the highest loss in glossiness, reaching up to 91.54%, while the black Chinese lacquer with 20% and 30% Fe(OH)_2_ additives exhibit the lowest loss in glossiness, ranging from 60.55% to 60.75%. This is due to the fact that when the amount of ferrous hydroxide is added insufficiently, the ferrous ion fails to coordinate with the phenolic hydroxyl group sufficiently, resulting in the inhibition of light aging of lacquer phenol by Fe(OH)_2_ is not obvious. At the same time, it also shows that Fe(OH)_2_ as an additive plays a protective role against ultraviolet aging. There is no significant difference in the proportion of glossiness reduction with different amounts of carbon black added to the Chinese lacquer.

### 3.2. Color Change

Chinese lacquer is a natural polymer that is highly sensitive to ultraviolet light, resulting in color changes [42]. The solidification of Chinese raw lacquer into a film is a complex biochemical process. Under the catalysis of lacquer enzyme, urushiol undergoes oxidation, and its products can absorb a large amount of visible light and ultraviolet light, leading to a darkening of the color [43]. After absorbing light energy, the lacquer film is prone to undergo further oxidation reactions, causing the formation of new products with reduced ability to absorb visible light, resulting in a lightening of the color. The addition of different pigments to Chinese lacquer can affect the film’s absorption of different light sources, thus influencing the color stability of Chinese lacquer [17]. The color of Chinese black lacquer is very close to pure black, with minimal changes in the a* and b* values during UV degradation. Therefore, this paper focuses on discussing the change in the lightness (*L*) value. Table 3 presents the lightness values of the two types of black Chinese lacquer before and during UV aging. The initial lightness values (*L*_0_) range from 8.88 to 9.24, indicating that the different additives and their amounts have little effect on the color. As the UV degradation time increases, the *L* value gradually increases, with the maximum value reaching 24.25, indicating that the color of black Chinese lacquer becomes progressively lighter.

Figure 2 illustrates the trend of color changes in black lacquers with different additives, which follows a similar pattern to the gloss loss rate. Both iron black lacquer and carbon black lacquer exhibit color difference changes during the first cycle (first 7 days). The color difference of iron black lacquer and carbon black lacquer increases with exposure time, with carbon black lacquer showing a faster rate of color difference change compared to iron black lacquer. There are certain differences in the color difference among the three formulations of iron black lacquer. In the first cycle, the color difference change in Fe10 is notably higher than that in Fe30, and Fe30 is higher than Fe20. This trend continues in the subsequent four cycles. The color difference change in Fe10 is significantly greater than that in Fe20 and Fe30, indicating that the amount of iron added is insufficient, and it also demonstrates that the addition of iron can inhibit the photodegradation of raw lacquer to some extent. Among the iron black lacquer group, Fe20 exhibits the smallest color difference. The Fe20 formulation with moderate additions seems to yield the best results in terms of minimizing color difference in the black lacquer. This is likely due to the fact that adding either too much or too little Fe(OH)_2_ can result in slower coordination reactions or incomplete reactions, both of which can impact the performance of the final black lacquer. After five cycles, the color difference of the iron black lacquer shows a gradual trend. The color difference in the carbon black lacquer group is not as pronounced as in the iron black lacquer group, and this pattern is similar to the glossiness results. The color difference of carbon black lacquer falls between Fe30 and Fe10, but the slope is steeper in the table, indicating that the color difference of carbon black lacquer still shows a noticeable upward trend. By the end of the fifth cycle, the color difference of C3 and C5 has surpassed that of Fe10. The difference in color among the three formulations of carbon black lacquer is not significantly different with respect to UV exposure time, indicating that the amount of carbon added has little effect on the color difference of carbon black lacquer. Therefore, from a color perspective, iron black lacquer demonstrates better color retention ability and antiaging effects compared to carbon black lacquer. Among the several formulations, Fe20 exhibits the smallest color difference, followed by Fe30.

### 3.3. SEM

Figure 3 presents the micromorphological changes in the black lacquer before and after UV exposure, captured at a magnification of 2000×. After 35 days of UV exposure, cracks appeared to varying degrees on the surface of both iron black lacquer and carbon black lacquer [44]. The surface of carbon black lacquer is uneven, displaying wrinkles and cracks. The iron black lacquer, when exposed to ultraviolet rays, exhibits a relatively flat surface with linear cracks, which may be influenced by ferrous hydroxide. In formulations with relatively low carbon and iron content, the surface unevenness of the black lacquer is evident, indicating that the addition of carbon and iron contributes to improving the surface quality of black lacquer. Additionally, black lacquer containing ferrous hydroxide demonstrates better film smoothness compared to black lacquer containing carbon black.

### 3.4. Infrared Spectroscopy Analysis

Infrared spectroscopy is a valuable method for studying the chemical changes occurring in Chinese lacquers after substances are added and also during UV degradation [10,45,46]. Figure 4 presents the infrared spectra of Chinese lacquer without additives and with the addition of different ratios of carbon black and Fe(OH)_2_. Comparing the infrared spectra of Chinese lacquer with and without the addition of Fe(OH)_2_, it can be observed that the absorption peak at 1690 cm^−1^ merges with the peak at 1600 cm^−1^ after the addition of Fe(OH)_2_. This indicates that a chemical reaction has occurred between ferrous hydroxide and Chinese lacquer, which is generally believed to be a coordination chemical reaction centered around the Fe^2+^ [31], resulting in the formation of compounds containing carboxyl groups, leading to the merging of these two peaks. Comparing the infrared spectra of Chinese lacquer with and without the addition of carbon black, their shapes are similar, suggesting that the likelihood of a chemical reaction occurring between carbon black and Chinese lacquer is low.

Figure 5 presents the spectra of six types of Chinese black lacquer before and during UV degradation. An evident broadband is observed in the spectra of both Chinese black lacquers, primarily dominant at 3300 cm^−1^, corresponding to the vibration mode νOH, which is associated with the hydroxyl group on the catechol ring [47]. This broadband can also be attributed to polysaccharide substances, although they exist in smaller quantities compared to catechol. The decline in the intensity of the broadband is primarily caused by the photochemical reaction of catechol. 

Upon UV light exposure, the shoulder at 3016 cm^−1^ assigned to the νCH=CH vibration of the sidechain diminishes, indicating the involvement of the sidechain in a photoinduced oxidation reaction. The reduction in intensity of the symmetric and asymmetric methylene peaks at 2920 cm^−1^ and 2855 cm^−1^, respectively, further suggesting intermolecular crosslinking resulting from sidechain oxidation reactions [47]. Additionally, the absorption peak at 1740 cm^−1^ shifts to 1710 cm^−1^, indicating the production of oxidation products [14,48]. However, after more than 21 days of UV degradation, the C=O (unsaturated) absorption peak at 1710 cm^−1^ slightly decreases, indicating further degradation of the resulting oxides. The decrease in the peak at 990 cm^−1^ is attributed to a decrease in conjugated trienes [27], indicating that with increased UV irradiation time, the urushiol side chains undergo oxidation, leading to an increase in diketones and other oxidation products.

To better analyze the effects of different dosages on the UV degradation of Chinese lacquer, the difference spectra of the two black Chinese lacquers are presented in Figure 6. The difference spectrum uses the infrared spectrogram after 35 days of degradation minus the pre-degradation spectrogram to obtain a value that more clearly reflects the changes in functional groups of the black lacquer before and after UV aging. The difference spectra exhibit several peaks with both negative and positive intensities, reflecting chemical changes occurring during UV degradation. The negative absorption in the difference spectrum indicates the formation of new structures during UV degradation, while the positive absorption indicates the loss of certain structures. The similarity in the difference spectra shapes between the two additives suggests a consistent UV degradation mechanism for black Chinese lacquer. This mechanism can be attributed to the oxidation of catechol and its sidechain unsaturated bonds. When comparing the effects of different concentrations of additives on black Chinese lacquer, it is observed that the difference in spectra is greatest for 10% Fe(OH)_2_ and 1% carbon black, indicating that the additives have an inhibitory effect on the UV degradation of black Chinese lacquer.

### 3.5. XPS Analysis

X-ray photoelectron spectroscopy (XPS) analysis is a highly effective method for investigating crucial information regarding the chemical environment and atomic concentration of materials. During the preliminary experiments, we evaluated two additives and six formulations, one of which contained 20% Fe(OH)_2_ and another with 3% carbon black addition. These particular formulations were chosen as they were deemed representative and demonstrated outstanding performance in terms of UV aging resistance. As a result, we selected these two formulations for further examination through XPS testing, facilitating a comparative study.

Figure 7a depicts the representative XPS survey spectra, while Table 4 presents the results of the XPS spectral analysis, including the mass fraction of carbon and oxygen atoms, along with the calculation of the carbon-to-oxygen ratio (C/O) for the tested samples.

The XPS spectra revealed that carbon (C) and oxygen (O) are the predominant elements, exhibiting characteristic peaks at binding energies of 284.91 eV and 532.91 eV, respectively [49]. Additionally, characteristic emission peaks at 396.91 eV and 709.8 eV were observed, indicating the presence of nitrogen and iron on the surface. The nitrogen signal is attributed to the presence of polysaccharides within the Chinese lacquer, while the iron signal originates from the Fe(OH)_2_ additive.

Analysis of the data presented in Table 4 demonstrates a decrease in the percentage of carbon atoms and a concomitant increase in the percentage of oxygen atoms in the UV-degraded samples compared to the control samples. This decrease in the carbon-to-oxygen ratio (C/O) from 3.50 and 4.22 for the Fe(OH)_2_ and carbon black additives, respectively, to 2.01 and 2.85, indicates that oxidation is the primary chemical reaction occurring during UV degradation of the Chinese black lacquer. The XPS analysis provides valuable insights into the chemical changes taking place in the Chinese black lacquer samples, highlighting the oxidation process and the associated alterations in the carbon and oxygen composition.

Figure 7b–e showcases the curve-fitting analysis of the C1s and O1s spectra for two Chinese black lacquer films. In the control lacquer film, it is evident that the peaks corresponding to the C=O functional groups in Fe20 are significantly lower compared to C3. This disparity can be attributed to the oxidation of Fe^2+^ to Fe^3+^, leading to an increased concentration of OH- ions. These OH- ions can then react with the C=O groups, resulting in the formation of stable products through the interaction of Fe ions with the Chinese lacquer. 

Table 5 presents the changes in carbon functional groups in Chinese lacquer with different additives before and after aging. Comparing the proportions of functional groups in the two different additives, it was found that the addition of Fe(OH)_2_ resulted in a higher proportion of C-OH and C-O-C groups compared to the addition of carbon black, while the CH group showed the opposite trend. This further confirms that a coordination reaction occurs between Fe(OH)_2_ and Chinese lacquer, leading to a reduction in CH groups and the formation of compounds containing C-OH and C-O-C groups. After UV degradation, the peaks associated with C-O-C and C-OH groups increased in the Fe20 sample, while the peak corresponding to CH groups decreased. This observation suggests that during the degradation process, the CH groups in the Fe(OH)_2_-coordinated product transformed into C-O-C and C-OH groups [50]. In the case of the carbon-added lacquer, it can be inferred that UV degradation caused damage to the C-H functional group, which subsequently oxidized to form C=O, C-O-C, and C-OH groups [20]. These findings are consistent with the results obtained from the infrared (IR) analysis.

The curve-fitting analysis of the C1s and O1s spectra provides valuable insights into the chemical changes occurring in the Chinese black lacquer films, particularly regarding the transformations and oxidation processes of various functional groups. These findings further support the observations made in the IR analysis, consolidating our understanding of the UV degradation mechanisms in the lacquer films.

## 4. Conclusions

In conclusion, this study investigated the ultraviolet degradation of black lacquer using ferrous hydroxide solution and carbon powder as additives. Different formulations of black lacquer were prepared with varying concentrations of ferrous hydroxide (10%, 20%, and 30%) and carbon black (1%, 3%, and 5%) added to natural raw lacquer. The analysis of the degradation process involved gloss measurement, color difference assessment, ambient scanning electron microscopy (SEM), Fourier infrared spectroscopy (FTIR), and X-ray photoelectron spectroscopy (XPS). These analytical techniques provided insights into the light aging mechanism of black lacquer and the impact of additives on its light aging performance, which holds great significance for the preservation of black lacquer cultural relics. The key findings of this research are as follows:

UV exposure led to a significant decrease in gloss value, an increase in color difference value, and the appearance of cracks in the electron microscope scanning images. Different formulations exhibited distinct effects of ferrous hydroxide and carbon powder on the light aging properties of raw lacquers. Notably, iron black lacquers demonstrated superior gloss retention and better resistance to light aging compared to carbon black lacquers.

Infrared spectroscopy confirmed the coordination reaction between ferrous hydroxide and raw lacquer, highlighting the formation of compounds containing carboxyl groups. Additionally, it indicated that carbon black and raw lacquer were physically mixed. With increased UV irradiation time, the side chains of lac phenols underwent oxidation, resulting in an increase in diketones and other oxidation products. Comparing the effects of different concentrations of additives on the black lacquer, it was found that the IR spectra of 10% Fe(OH)_2_ and 1% carbon black had the greatest difference, indicating that the additives inhibited the UV degradation of the black lacquer.

The post-UV degradation samples exhibited a decrease in the ratio of carbon atoms and an increase in the ratio of oxygen atoms, leading to the destruction of C-H functional groups, which subsequently oxidized to C=O. After the addition of Fe(OH)_2_ and carbon black, the C/O ratios of the black lacquers decreased, showing an inverse relationship with the aging time of UV exposure. This highlighted oxidation as the primary chemical reaction during the UV degradation process. Moreover, XPS curve fitting analysis provided quantitative data on the percentage of functional groups in the varnish film before and after aging, further enhancing our understanding of the changes occurring in black lacquer during aging.

In summary, this research offers valuable insights into the light aging mechanisms of black lacquer and underscores the advantages of using ferrous hydroxide and carbon black additives to enhance its stability and preservation. The findings contribute to the existing body of knowledge on black lacquer conservation and present practical implications for safeguarding and protecting black lacquer cultural relics.

## Figures and Tables

**Figure 1 materials-16-05607-f001:**
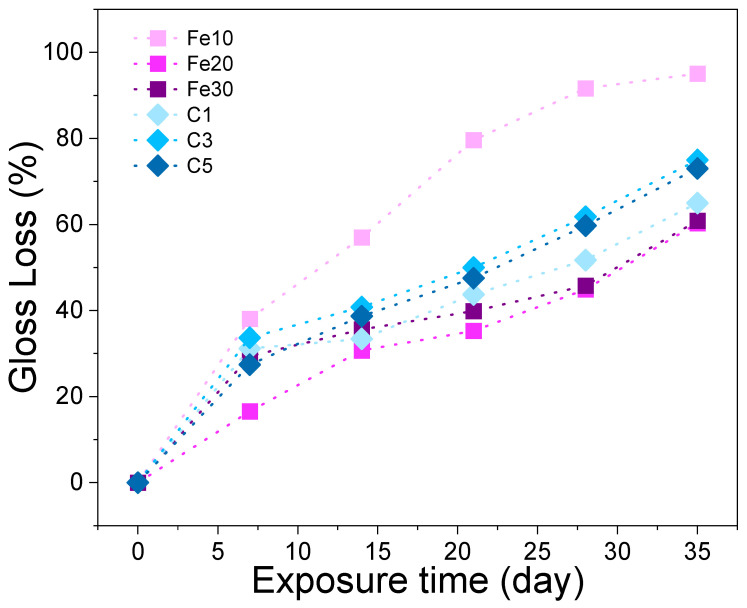
The relationship between UV exposure time and gloss loss rate.

**Figure 2 materials-16-05607-f002:**
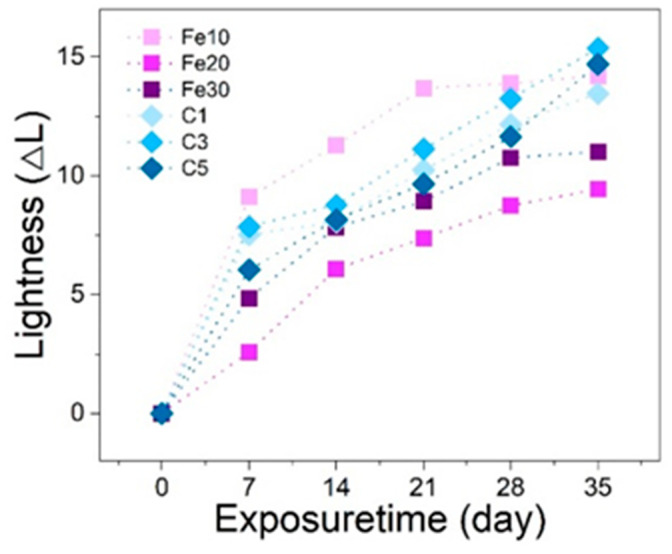
The relationship between UV exposure time and lightness difference.

**Figure 3 materials-16-05607-f003:**
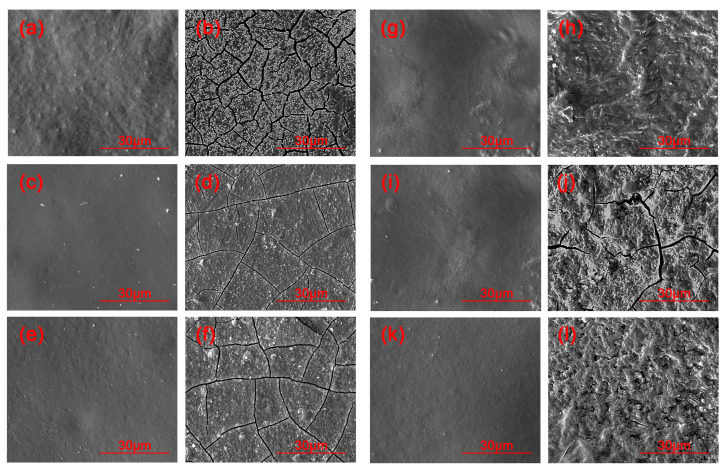
SEM images of black lacquer before and after UV exposure. (**a**) C1 before aging; (**b**) C1 after aging; (**c**) C3 before aging; (**d**) C3 after aging; (**e**) C5 before aging; (**f**) C5 after aging; (**g**) Fe10 before aging; (**h**) Fe10 after aging; (**i**) Fe20 before aging; (**j**) Fe20 after aging; (**k**) Fe30 before aging; (**l**) Fe30 after aging.

**Figure 4 materials-16-05607-f004:**
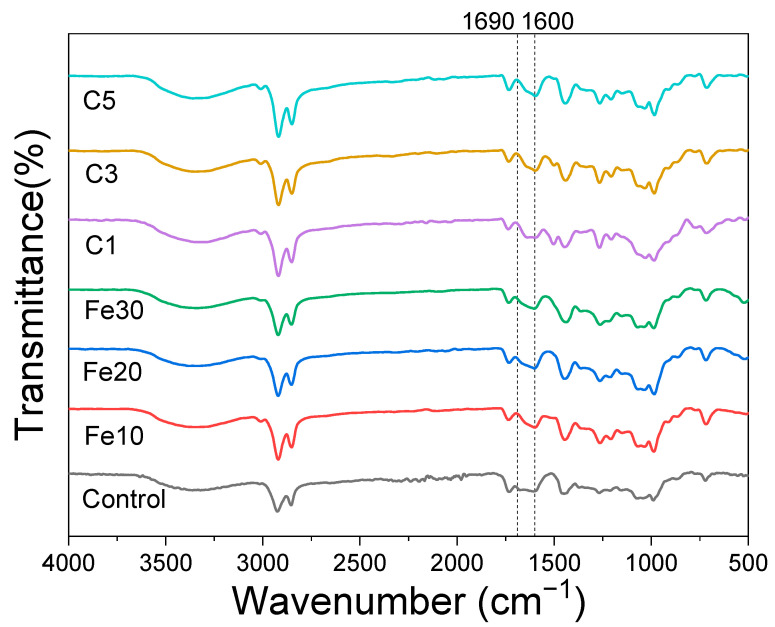
Infrared spectra of raw lacquers, iron black lacquers, and carbon black lacquers before UV degradation.

**Figure 5 materials-16-05607-f005:**
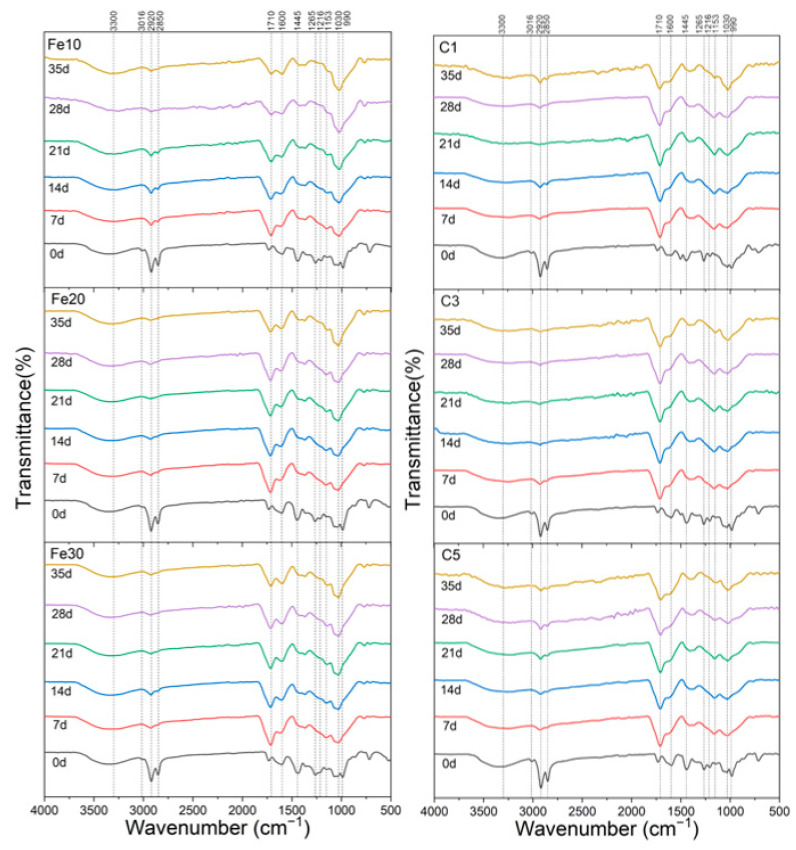
IR spectra of two types of Chinese black lacquer before and during UV degradation: Fe(OH)_2_ added (**left**) and carbon black added (**right**).

**Figure 6 materials-16-05607-f006:**
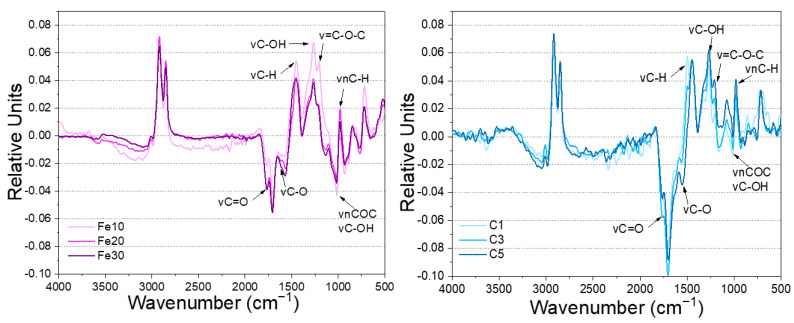
IR difference spectra of two types of Chinese black lacquer: Fe(OH)_2_ added (**left**) and carbon black added (**right**).

**Figure 7 materials-16-05607-f007:**
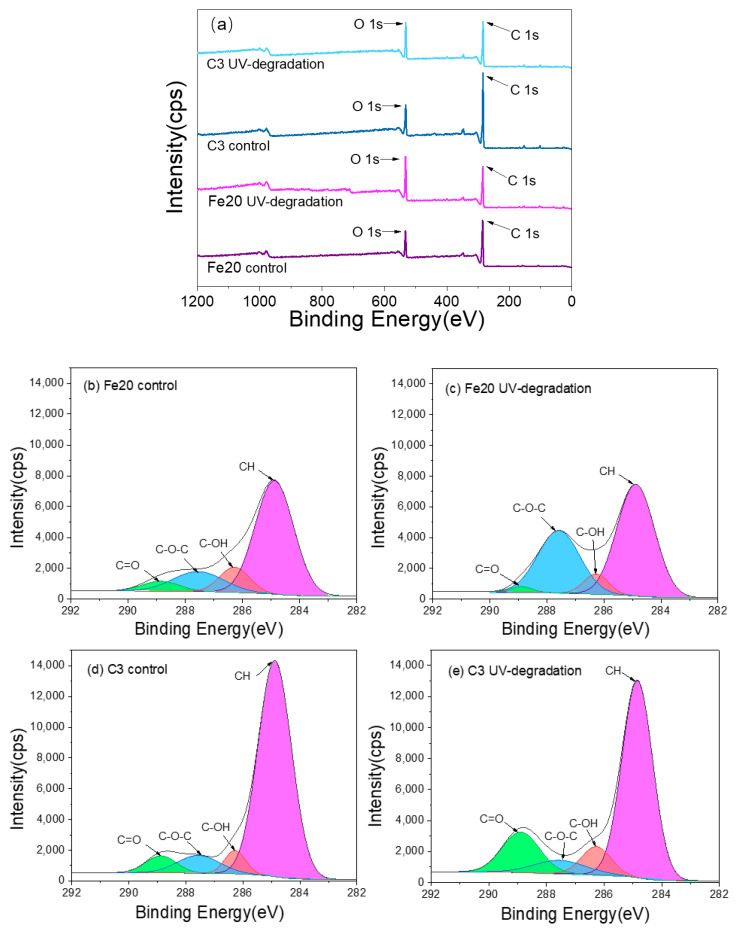
XPS survey spectra of two types of Chinese black lacquer before and after UV degradation. (**a**) X-ray photoelectron spectroscopy survey spectra of C1s and O1s. (**b**–**e**) Curve-fitting of the C1s spectra of the black lacquer.

**Table 1 materials-16-05607-t001:** Black lacquer specimens with different additives.

Code	Additives	Number of Specimens
L-Fe10	10% Fe(OH)_2_	8
L-Fe20	20% Fe(OH)_2_	8
L-Fe30	30% Fe(OH)_2_	8
L-C1	1% carbon black	8
L-C3	3% carbon black	8
L-C5	5% carbon black	8

**Table 2 materials-16-05607-t002:** Gloss of black lacquer before and during UV aging.

Code	H_0_ (%)	H_7_ (%)	H_14_ (%)	H_21_ (%)	H_28_ (%)	H_35_ (%)
Fe10	21.4 ^c^	13.3 ^b^	9.3 ^c^	4.4 ^c^	1.8 ^c^	1.8 ^c^
Fe20	32.9 ^b^	27.4 ^a^	22.7 ^b^	21.3 ^b^	18.1 ^b^	13.0 ^b^
Fe30	47.2 ^a^	33.4 ^a^	30.4 ^a^	28.4 ^a^	25.6 ^a^	18.5 ^a^
*p* values	<0.0001	<0.0001	<0.0001	<0.0001	<0.0001	<0.0001
C1	34.7 ^a^	23.7 ^a^	23.0 ^a^	19.5 ^a^	16.7 ^a^	12.1 ^a^
C3	29.6 ^b^	19.7 ^b^	17.6 ^b^	14.9 ^b^	11.4 ^b^	7.4 ^b^
C5	28.7 ^b^	20.8 ^b^	17.5 ^b^	14.9 ^b^	11.4 ^b^	7.7 ^b^
*p* values	<0.0001	<0.0001	<0.0001	<0.0001	<0.0001	<0.0001

Note: Mean values of H followed by the same small superscript letters (^a–c^) within a group are not significantly different based on Fisher’s Protected LSD test at the 0.05 significance level. The *p*-value indicates the significance of the influencing factor. A smaller *p*-value indicates stronger significance. Generally, when the *p*-value is less than 0.05, the influencing factors are considered significant, and when the *p*-value is greater than 0.05, they are considered insignificant.

**Table 3 materials-16-05607-t003:** Color lightness difference before and after UV aging of black lacquer.

Code	L_0_	L_7_	L_14_	L_21_	L_28_	L_35_
Fe10	9.15 ^a^	18.24 ^a^	20.42 ^a^	22.83 ^a^	23.02 ^a^	23.34 ^a^
Fe20	9.14 ^a^	11.71 ^c^	15.21 ^b^	16.51 ^b^	17.88 ^c^	18.56 ^b^
Fe30	9.13 ^a^	13.98 ^b^	16.98 ^b^	17.03 ^b^	19.89 ^b^	20.16 ^b^
*p* values	0.910	<0.0001	<0.0001	<0.0001	<0.0001	<0.0001
C1	9.02 ^a^	14.95 ^a^	16.54 ^a^	18.50 ^a^	21.18 ^a^	22.48 ^a^
C3	8.88 ^a^	16.74 ^a^	16.90 ^a^	20.01 ^a^	22.13 ^a^	24.25 ^a^
C5	9.24 ^a^	15.29 ^a^	17.41 ^a^	18.01 ^a^	20.88 ^a^	23.94 ^a^
*p* values	0.755	0.776	0.406	0.527	0.734	0.522

Note: Mean values of H followed by the same small superscript letters (^a–c^) within a group are not significantly different based on Fisher’s protected LSD test at the 0.05 significance level. The *p*-value indicates the significance of the influencing factor. A smaller *p*-value indicates stronger significance. Generally, when the *p*-value is less than 0.05, the influencing factors are considered significant, and when the *p*-value is greater than 0.05, they are considered insignificant.

**Table 4 materials-16-05607-t004:** Experimental carbon and oxygen atom composition and C/O ratio obtained using XPS analysis for measured samples.

Code	%C	%O	C/O
Fe20 control	77.77	22.23	3.50
Fe20 UV degradation	62.58	31.07	2.01
C3 control	80.83	19.17	4.22
C3 UV degradation	72.88	25.58	2.85

**Table 5 materials-16-05607-t005:** Percentage of functional groups of lacquer film before and after aging of black lacquer.

Code	CH	C-OH	C-O-C	C=O
Fe20 control	66.86	11.68	15.73	5.73
Fe20 UV degradation	56.70	6.75	35.11	1.45
C3 control	80.47	5.37	8.38	5.79
C3 UV degradation	66.77	8.97	7.53	16.74

## Data Availability

Not applicable.

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
