# Peer review of "Comparative Study on UV Degradation of Black Chinese Lacquers with Different Additives"

_materials, 2023, doi:10.3390/ma16165607_

Round 1
Reviewer 1 Report
The paper entitled Comparative study on UV-degradation of black Chinese lacquers with different additives' is worth studying. However, some issues must be addressed.
(1) The addition of carbon black and ferrous hydroxide to the lacquer does not seem to have strong reasons. We should also add the current method to slow the decolorization of black lacquer.
(2) The amount of added substance i.e. carbon black and ferrous hydroxide does not seem to have good bases (such as mole ratio).
(3) The last paragraph of the Introduction needs references.
(4) Fig 4 The transmittance (y-axis) may be written as an arbitrary unit.
(5) Related to the data in Table 5, a better explanation and possible chemical reaction is needed.
(6) Fig 6 and Fig 7 should be combined for clarity.
(7) Is there any known chemical reaction related to the color changes of the lacquers?
(8) Some chemical formulae are not well-written, such as the caption of Fig. 4. Check the rest.
English is sound, but needs typo check.
Reviewer 2 Report
The authors report the UV-degradation of black Chinese lacquer by incorporating carbon black and ferrous hydroxide as additives. The main methods employed for analysis were gloss loss measurement, color change assessment, SEM imaging, FTIR spectroscopy, and XPS analysis. The research employed various analytical techniques to examine the effects of carbon black and ferrous hydroxide additives on the lacquer's properties. The addition of additives demonstrated a noticeable impact on some studied properties.
In my opinion the text should explore new and valuable methodologies to protect the material against solar radiation. However, the text it is well written, the methodologies and the plannification of the experiments are well design, so I recommend accept the article in present form however I also should desirable to correct the typo mistakes
Reviewer 3 Report
Authors study the degradation of black Chinese lacquers with additives. I believe that many of the results presented should be compared with the Chinese lacquer without additives. Additionally, there are several sections of the manuscript that need to be revised.
The composition of each sample should be analyzed using elemental analysis and X-ray Fluorescence or ICP-MS.
I don't fully understand the method used for applying the Fe or carbon additives to the lacquer.
What is the gloss level without adding the different additives? It would be beneficial to include control measurements to observe the improvement in the process.
Figure 1 shows the loss of gloss over time, and all profiles exhibit the same trend except for Fe10. Can this be explained?
What is the precision of the results shown in tables 2 and 3, as they display two decimal places? Does the method have a precision of 0.01% error? If not, rounding the results would make them easier to understand.
Why are there color differences, with Fe10 < Fe30 < Fe20? It doesn't follow a logical order, can this be explained?
In Figure 3, the effects of aging are analyzed using SEM. Why does Figure 3g appear to be the least affected, with fewer cracks compared to other cases? Also, the SEM images have excessive brightness, making them difficult to interpret. Could they be homogenized for better visualization?
To observe the effects of additives, a blank sample (without additives) should be analyzed using IR, as the observed effects are primarily related to the lacquer base, not the additives.
Additionally, in Figure 5, which bands correspond to "before degradation"?
What do A and B represent in Figure 6? A: Control and B: Degradation?
The XPS analysis has been performed only on one sample each of Fe and C. Why haven't the rest of the samples been analyzed?
Considering the heterogeneity of the results, I am unsure if this article fulfills its objective. I suggest rewriting the conclusions to highlight the advantages provided in comparison to the previous research.
Lastly, I believe that this manuscript requires major revisions before being considered for publication.
Round 2
Reviewer 1 Report
The revised form is much better. The concerns raised by the reviewers have been addressed. I would accept the current version.
The English is OK.
Reviewer 3 Report
In this paper, the authors the degradation of black Chinese lacquers with additives. The content of this article is appropriate due to the results shown. Also, the authors have responsed to all my questions. Therefore, I consider that the paper is suitable for publication in present form.